# Differences in Cerebral Metabolism between Moderate- and High-Severity Groups of Patients with Out-of-Hospital Cardiac Arrest Undergoing Target Temperature Management

**DOI:** 10.3390/brainsci13101373

**Published:** 2023-09-26

**Authors:** Yeonho You, Changshin Kang, Wonjoon Jeong, Hong Joon Ahn, Jung Soo Park, Jin Hong Min, Yong Nam In, Jae Kwang Lee, So Young Jeon

**Affiliations:** 1Department of Emergency Medicine, Chungnam National University Hospital, 282 Munhwa-ro, Jung-gu, Daejeon 35015, Republic of Korea; yyo1003@naver.com (Y.Y.); gardenjun@hanmail.net (W.J.); jooniahn@daum.net (H.J.A.); cpcr@cnu.ac.kr (J.S.P.); chloe9899@naver.com (S.Y.J.); 2Department of Emergency Medicine, College of Medicine, Chungnam National University, 282 Mokdong-ro, Jung-gu, Daejeon 35015, Republic of Korea; laphir2006@naver.com (J.H.M.); ynsoft@naver.com (Y.N.I.); 3Department of Emergency Medicine, Chungnam National University Sejong Hospital, 20 Bodeum 7-ro, Sejong 30099, Republic of Korea; 4Department of Emergency Medicine, Konyang University Hospital, College of Medicine, Daejeon 35365, Republic of Korea

**Keywords:** cardiac arrest, lactic acid, prognosis

## Abstract

The aim of this study was to investigate the differences in cerebral metabolism and the prognostic value of cerebrospinal fluid (CSF) lactate 24 h after the return of spontaneous circulation (ROSC) in patients with out-of-hospital cardiac arrest (OHCA). CSF lactate and pyruvate levels were measured immediately and every 2 h for 24 h after the ROSC. The distribution of cerebral mitochondrial dysfunction (MD) and cerebral ischemia was also evaluated. In the moderate-severity group, the absence of cerebral MD or ischemia was observed in six patients (40.0%) immediately after ROSC and in nine patients (60.0%) 24 h after the ROSC. In the high-severity group, the absence of cerebral MD or ischemia was observed in four patients (30.8%) immediately after ROSC and in three patients (23.1%) 24 h after the ROSC. The distribution of cerebral metabolism over time varied depending on the severity of the OHCA. The predictive value of CSF lactate levels for a poor neurological prognosis was better for patients in the moderate-severity group than for the overall patient cohort. Therefore, the severity in the patients with OHCA should be considered when studying cerebral metabolism or using CSF lactate as a prognostic tool.

## 1. Introduction

An increase in the lactate/pyruvate (LP) ratio, which differs in cerebral ischemia and mitochondrial dysfunction (MD), occurs in patients with cardiac arrest (CA) [1,2]. In cerebral ischemia, when the cerebral blood flow is interrupted, pyruvate decreases due to decreased oxygen and glucose supply, which is necessary for energy metabolism, leading to an elevated LP ratio. In MD, the LP ratio increases but the pyruvate level is normal or increased due to an increase in the glycolysis rate [3,4,5]. An animal study found that treatment for MD occurring after CA is distinct from ischemia-related therapies, and the use of nitrite therapy as a form of treatment for MD was reported [6].

Lactate is a predictor of a poor neurological prognosis in patients with CA undergoing target temperature management (TTM) [7]. Previous studies reported that cerebrospinal fluid (CSF) lactate levels show better performance in predicting a poor neurological prognosis than serum lactate levels, and that CSF lactate measured at 24 h after the return of spontaneous circulation (ROSC) has the highest prognostic performance [8,9]. In patients with a poor neurological prognosis, CSF lactate and pyruvate levels increase during the first 24 h after ROSC and remain elevated for up to 48 h. However, the CSF LP ratio is elevated (≥16) during the first 20 h after ROSC regardless of the neurological prognosis, and then it normalizes [10].

The neurological prognosis of patients with CA is influenced by various factors, but regardless of the post-cardiac-arrest care, the CA severity is a significant factor in the final prognosis, either a good or poor outcome. A previous study reported that the revised post-cardiac-arrest syndrome for therapeutic hypothermia (rCAST) score is useful for distinguishing CA severity [11]. However, no studies have examined the differences in cerebral metabolism and the prognostic value of CSF lactate based on CA severity in patients with out-of-hospital cardiac arrest (OHCA). Therefore, in this study, we aimed to investigate the differences in cerebral metabolism and the prognostic value of CSF lactate during the 24 h after the ROSC in patients with OHCA based on CA severity using the rCAST score.

## 2. Materials and Methods

### 2.1. Ethical Approval and Consent

This study was approved by the Institutional Review Board of Chungnation National University Medical Centre (CNUH IRB 201907033003). All procedures and protocols were performed according to the Declaration of Helsinki and the International Conference of Harmonization and Good Clinical Practice and are reported according to the CONSORT criteria. Written informed consent, approval for the donation of human materials, and research on human medical ethics were obtained from the patients’ next of kin.

### 2.2. Study Design and Patients

This prospective observational cohort study was conducted at a single center and included patients with OHCA who underwent TTM between July 2021 and June 2023. The primary endpoint was to evaluate whether a difference exists in the pattern of cerebral MD and ischemic occurrence over time based on CA severity using the rCAST score. For the secondary endpoint, we investigated the difference in the prognostic value of CSF lactate during the 24 h after ROSC based on CA severity using the rCAST score. The neurological prognosis was assessed at 6 months after ROSC using the Glasgow–Pittsburgh Cerebral Performance Categories (CPCs) scale, through either face-to-face interviews or structured telephone interviews [12], as follows: CPC 1 (good performance), CPC 2 (moderate disability), CPC 3 (severe disability), CPC 4 (vegetative state), or CPC 5 (brain death or death) [13]. Telephone interviews were conducted by an emergency physician who was fully informed about the protocol. Cerebral Performance Categories (CPCs) 1 and 2 were defined as having a good neurological prognosis and CPCs 3–5 as having a poor neurological prognosis. Enrolled patients were classified based on CA severity scores: low severity (rCAST ≤ 5.5), moderate severity (rCAST 6–14), and high severity (14.5–18.5) [11]. We included patients with OHCA who underwent TTM with a Glasgow Coma Scale (GCS) score ≤ 8 after ROSC. The exclusion criteria were as follows: (1) age < 18 years; (2) CA due to trauma; (3) TTM discontinued due to hemodynamic instability; (4) TTM not performed due to cerebral hemorrhage, active bleeding, poor neurologic status prior to CA, or known terminal illness; (5) severe cerebral edema, obliteration of the basal cisterns, or an intracranial mass on brain computed tomography; (6) ineligibility for lumbar puncture due to antiplatelet therapy, anticoagulation therapy, coagulopathy, platelet count < 40,000/mL, or international normalized ratio > 1.5 [14]; (7) extracorporeal membrane oxygenation provided; and (8) consent for lumbar puncture or other procedures not granted by the next of kin.

### 2.3. TTM Protocol

Core body temperature was monitored using an esophageal and bladder temperature probe. TTM was maintained for 24 h after achieving the target temperature of 33 °C, followed by rewarming to 37 °C at a rate of 0.25 °C/h using cooling devices (Arctic Sun^®^ Energy Transfer Pads^TM^; Medivance Corp., Louisville, KY, USA). Midazolam (0.05 mg/kg intravenous bolus, followed by a titrated intravenous continuous infusion at a dose between 0.05 and 0.2 mg/kg/h) and rocuronium (0.6 mg/kg intravenous bolus, followed by an infusion of up to 0.5 mg/kg/h) were administered for sedation and to control shivering. All patients were monitored using a continuous amplitude-integrated electroencephalogram (aEEG; Olympic Medical CFM 6000; Natus Inc., Seattle, WA, USA). Antiepileptic drugs were administered (levetiracetam: loading dose, 2 g bolus intravenously; maintenance dose, 1 g bolus twice daily intravenously) when evidence of electrographic seizure was present, or a clinical diagnosis of a seizure was made. Fluid resuscitation or vasopressors were administered when necessary to maintain a mean arterial pressure of between 65 and 100 mmHg [15].

### 2.4. Data Collection

The following data were retrieved from the database: age; sex; presence of a witness at the time of the collapse; cardiopulmonary resuscitation (CPR) administered by the bystander; cardiac rhythm during first monitoring; etiology of CA; time from collapse to CPR (no flow time); time from CPR to ROSC (low-flow time); time from ROSC to the achievement of a target temperature of 33 °C (induction time); time from ROSC to insertion of the lumbar catheter (lumbar puncture time); arterial partial pressure of oxygen; CSF partial pressure of oxygen; serum glucose, CSF glucose, aspartate transaminase, alanine transferase, total bilirubin, and neutrophil gelatinase-associated lipocalin measured immediately and 24 h after ROSC; Charlson Comorbidity Index GCS scores immediately after ROSC; pH and lactate levels obtained immediately after ROSC; and CPC at 6 months after ROSC. aEEG findings were recorded immediately and every 1 h for 24 h after ROSC.

### 2.5. Measurement of Lactate, Pyruvate, and LP Ratio

A lumbar catheter was inserted using a Hermetic^TM^ lumbar accessory kit (Integra Neurosciences, Plainsboro, NJ, USA) between the third and fourth lumbar vertebrae of the patient lying in the lateral decubitus position with the hips and knees flexed. Serum was obtained using a radial arterial catheter. CSF was collected in a perchloric acid tube and centrifuged at 3000 rpm for 10 min. The supernatants obtained after centrifugation (from both serum and CSF containing tubes) were immediately frozen and stored at −40 °C until analysis. CSF lactate and pyruvate levels were immediately measured and every 2 h for 24 h after ROSC. An electrochemiluminescence immunoassay kit (COBAS^®^ e801; Roche Diagnostics, Rotkreuz, Switzerland) was used to automatically discard aliquots with hemolysis levels exceeding a defined threshold. The CSF LP ratio was calculated as CSF lactate level divided by the CSF pyruvate level [16]. Cerebral MD was defined as a CSF LP ratio > 16 and a CSF pyruvate level > 0.07 mmol/L, and cerebral ischemia was defined as a CSF LP ratio > 16 and a CSF pyruvate level ≤ 0.07 mmol/L [10,17].

### 2.6. Sample Size

Based on a previous study [8], the area under the receiver operating characteristic curve for predicting a poor neurological prognosis using CSF lactate levels measured 24 h after ROSC was 0.89 in patients with OHCA treated with TTM. Thirty patients were required to achieve a power of 0.99 at a significance level of 0.05 (two-sided test), considering a 10% missing rate.

### 2.7. Statistical Analysis

Continuous variables are reported as medians with interquartile ranges or means and standard deviations, depending on the normal distribution. Categorical variables are reported as frequencies and percentages. Comparisons between groups were performed using the chi-square test, Fisher’s exact test, Mann–Whitney *U* test, independent *t*-test, Kruskal–Wallis test, or ANOVA test. The area under the receiver operating characteristic curve was used to identify the prognostic value of lactate, pyruvate, and LP ratio for predicting the neurological prognosis. All statistical analyses were performed using PASW/SPSS software version 18 (IBM, Armonk, NY, USA) and MedCalc 15.2.2 (MedCalc Software, Mariakerke, Belgium). Statistical significance was set at *p* < 0.05 (two-tailed).

## 3. Results

### 3.1. Patient Characteristics

Thirty patients were enrolled in the study. Of these, 17 patients died, all of whom had a CPC of 5; 14 patients died due to pneumonia; and organ donation was performed after brain death in 3 patients. The low-, moderate-, and high-severity groups comprised 2 (6.7%), 15 (50.0%), and 13 (43.3%) patients, respectively (Figure 1). Among the enrolled patients, 9 (30.0%), 0 (0.0%), 1 (3.3%), 3 (10.0%), and 17 (56.7%) had CPC values of 1, 2, 3, 4, and 5, respectively. One patient showed waves indicative of a burst suppression pattern on their aEEG at 15 h after ROSC. Throughout the research period, 32 patients with low severity were excluded from the study because they demonstrated conscious recovery with a GCS score of ≥9 after ROSC. Only two patients with low-severity CA were included in this study. Therefore, comparisons based on the severity were conducted between the moderate- and high-severity groups.

In the moderate-severity group, 9 (60.0%) cases showed a poor neurological prognosis, while in the high-severity group, 12 (92.3%) cases showed a poor neurological prognosis (*p* = 0.02) (Table 1).

### 3.2. Prognostic Value of CSF Lactate, Pyruvate, and Lactate/Pyruvate Ratio

The two patients with low-severity CA had a good neurological prognosis. Among the 13 patients with high-severity CA, only 1 demonstrated a good neurological prognosis. Because of this distribution, a comparative analysis of the prognostic value of CSF lactate based on severity was not feasible. Therefore, the prognostic value of CSF lactate levels was compared between the overall patient group and the moderate-severity group. The value of CSF lactate in predicting a poor neurological prognosis differed between the overall patient and the moderate-severity groups, with the CSF lactate level demonstrating superior performance in the moderate-severity group (Table 2).

### 3.3. Distribution of Cerebral Ischemia and MD

Among the 28 patients (the 2 patients with low-severity CA were excluded), 10 (35.7%) did not present with MD or ischemia due to cerebral metabolism immediately after ROSC. Among these patients, seven (70.0%) had a poor neurological prognosis. Cerebral MD was the most prevalent metabolic pattern immediately after ROSC, occurring in 17 (60.7%) patients. Of these, 13 (76.5%) had a poor neurological prognosis. One patient (3.6%) demonstrated a cerebral ischemic pattern immediately after ROSC and had a poor neurological prognosis. Among the four patients who consistently displayed cerebral MD up to 24 h after ROSC, all had a poor neurological prognosis. A total of 12 (42.9%) of the 28 patients did not show cerebral MD or ischemia as cerebral metabolism 24 h after ROSC, with 6 of these patients (50.0%) experiencing a poor neurological prognosis. Cerebral MD was observed in 10 patients (35.7%) 24 h after ROSC, and all of these patients had a poor neurological prognosis. Six patients (21.4%) presented with cerebral ischemia 24 h after ROSC, and five (83.3%) of these had a poor neurological prognosis.

### 3.4. Distribution of Cerebral Ischemia and MD Immediately after ROSC Based on CA Severity

For the two patients with low-severity C, one patient did not show cerebral MD or ischemia; the other patient who presented with cerebral MD immediately after ROSC had a good neurological prognosis. No significant differences were found in the prevalence of cerebral MD or ischemia between the moderate- and high-severity groups immediately after ROSC (*p* = 0.81). In the moderate-severity group, six patients (40.0%) did not show cerebral MD or ischemia immediately after ROSC. Of these, three patients (50.0%) had a poor neurological prognosis. Of the eight moderate-severity patients (53.3%) who presented with cerebral MD immediately after ROSC, five (62.5%) had a poor neurological prognosis. The one moderate-severity patient (6.7%) who exhibited cerebral ischemia immediately after ROSC had a poor neurological prognosis. In the high-severity group, four patients (30.8%) without cerebral MD or ischemia immediately after ROSC had a poor neurological prognosis. Among the nine high-severity patients (69.2%) with cerebral MD immediately after ROSC, eight (88.9%) had a poor neurological prognosis. None of the high-severity patients showed cerebral ischemia immediately after ROSC (Table 3).

### 3.5. Distribution of Cerebral Ischemia and MD 24 h after ROSC Based on Severity

In the low-severity group, none of the patients presented with cerebral MD or ischemia 24 h after ROSC. In the moderate-severity group, nine patients (60.0%) did not show cerebral MD or ischemia 24 h after ROSC. Of these, three patients (33.3%) had a poor neurological prognosis. Five patients (33.3%) presented with cerebral MD and one patient (6.7%) had cerebral ischemia 24 h after ROSC; these patients had a poor neurological prognosis. In the high-severity group, three patients (23.1%) without cerebral MD or ischemia 24 h after ROSC had a poor neurological prognosis. Additionally, five patients (38.5%) with cerebral MD 24 h after ROSC had a poor neurological prognosis. Of the five patients (38.5%) with cerebral ischemia 24 h after ROSC, four (80.0%) had a poor neurological prognosis. Consequently, the high-severity group demonstrated a higher frequency of cerebral MD or ischemia than the moderate-severity group 24 h after ROSC (*p* = 0.04) (Table 3, Figure 2).

## 4. Discussion

This study is the first to evaluate the differences in cerebral metabolism and the prognostic value of CSF lactate in the 24 h after ROSC in patients with OHCA based on CA severity using the rCAST score. Patients with high disease severity demonstrated a higher frequency of cerebral MD or ischemia than those with moderate disease severity. In particular, patients who continued to exhibit cerebral MD for up to 24 h after ROSC had a poor neurological prognosis. Additionally, the predictive value of CSF lactate levels for determining a poor neurological prognosis demonstrated better performance for patients with moderate-severity CA than for the overall patient cohort. These findings suggest that the skewed distribution of the neurological prognosis into either a good or poor neurological prognosis among patients with OHCA with low- or high-severity CA could have led to these results. Therefore, differentiating and using CSF lactate levels as a prognostic tool based on OHCA severity may be essential. In previous studies, parameters such as shockable rhythm, witnessed CA, no-flow time, low-flow time, cardiac etiology, and age were associated with the neurological prognosis of OHCA patients [18]. However, in this study, the majority of patients exhibiting low-severity CA according to the rCAST score demonstrated conscious recovery after ROSC and were excluded from the study. Consequently, apart from the witnessed CA, there were no significant between-group differences in the other variables.

As lactate in the ionic state passes slowly through the blood–brain barrier, CSF lactate is reflected in the anaerobic metabolism of cerebral glycolysis independent of serum lactate levels. Therefore, the lactate concentration in the CSF is similar to that in the brain, whereas the lactate level in the jugular vein is always closer to that in the arterial blood and often differs from the CSF lactate level [19,20]. The lactate metabolized in the liver and kidneys might increase due to infection, seizures, mitochondrial disorders, and ischemia [21,22,23]. Additionally, red blood cells contain a high amount of lactate; hence, hemolysis can increase the lactate levels [24,25]. In this study, hemolyzed samples exceeding a defined threshold value were discarded. Differences were not found in aspartate transaminase, alanine transferase, total bilirubin, and neutrophil gelatinase-associated lipocalin levels measured immediately and 24 h after ROSC between the moderate- and high-severity groups.

Cerebral MD, characterized by normal or elevated cerebral pyruvate and elevated cerebral lactate, leads to disruption of the blood–brain barrier and apoptotic cell death through the release of cytochrome c, opening of mitochondrial permeability transition pores, calcium accumulation, and increased ROS production [4,26,27]. In this study, patients with high-severity CA demonstrated a higher frequency of cerebral MD or ischemia and had a worse neurological prognosis than those with moderate-severity CA. In previous decades, lumbar CSF drainage was contraindicated because of the risk of transtentorial or tonsillar herniation when the intracranial pressure increased. However, recent studies have shown that intracranial pressure rarely exceeds 20 mmHg in OHCA patients undergoing TTM, and that lumbar CSF drainage, when performed with careful monitoring of the intracranial pressure and vital signs, might be a valuable treatment option for managing refractory increased intracranial pressure in patients with discernible basal cisterns [28,29].

This study had several limitations. First, this was a single-center study with a small sample size. However, in a previous study, the AUROC for CSF lactate levels measured 24 h after ROSC for the prediction of a poor neurological prognosis was 0.89, and 30 patients were required to achieve a power of 0.99 at a significant level of 0.05, considering a 10% missing rate. Additionally, a post hoc sample size calculation revealed that AUROC for CSF lactate levels measured 24 h after ROSC for prediction of a poor neurologic prognosis was 0.90, and that 22 patients were required to achieve a power of 0.99 at a significance level of 0.05. Second, CSF glucose levels and CSF partial pressure of oxygen were not continuously assessed. However, no differences were noted between the moderate- and high-severity groups in the study when CSF glucose levels and CSF partial pressure of oxygen were evaluated immediately and 24 h after ROSC. Third, the parameters were measured for 24 h after ROSC; thus, changes in the parameters over a longer duration could not be determined. Fourth, other biomarkers of blood–brain barrier disruption, such as S100B and neuron-specific enolases, were not measured. Fifth, age-related changes in the CSF flow were not considered. Finally, the investigator was not blinded to the experimental conditions. Therefore, future studies with blinded investigators are necessary to overcome these limitations.

## 5. Conclusions

The distribution of cerebral metabolism over time varied according to the severity of OHCA, and patients with high-severity CA demonstrated a higher frequency of cerebral MD or ischemia. Additionally, the predictive value of CSF lactate levels for a poor neurological prognosis was better for patients with moderate-severity CA than for the overall patient cohort. Therefore, patient CA severity should be considered when studying cerebral metabolism or using CSF lactate as a prognostic tool.

## Figures and Tables

**Figure 1 brainsci-13-01373-f001:**
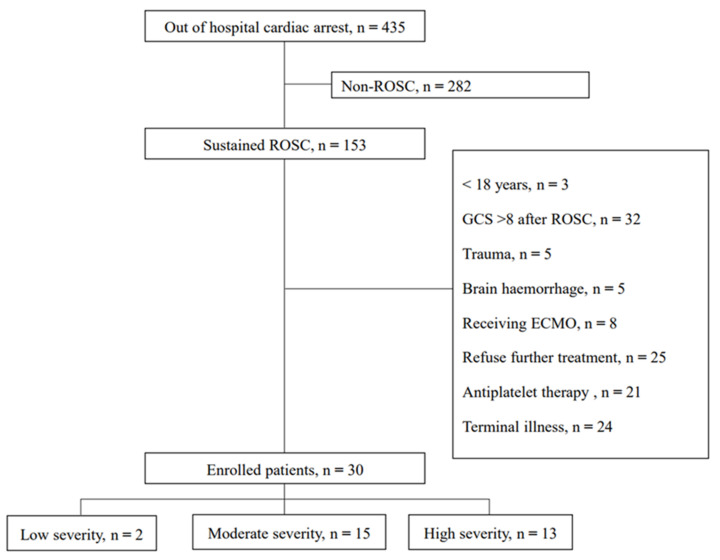
Flow diagram of included patients. Abbreviations: ROSC, return of spontaneous circulation; ECMO, extracorporeal membrane oxygenation.

**Figure 2 brainsci-13-01373-f002:**
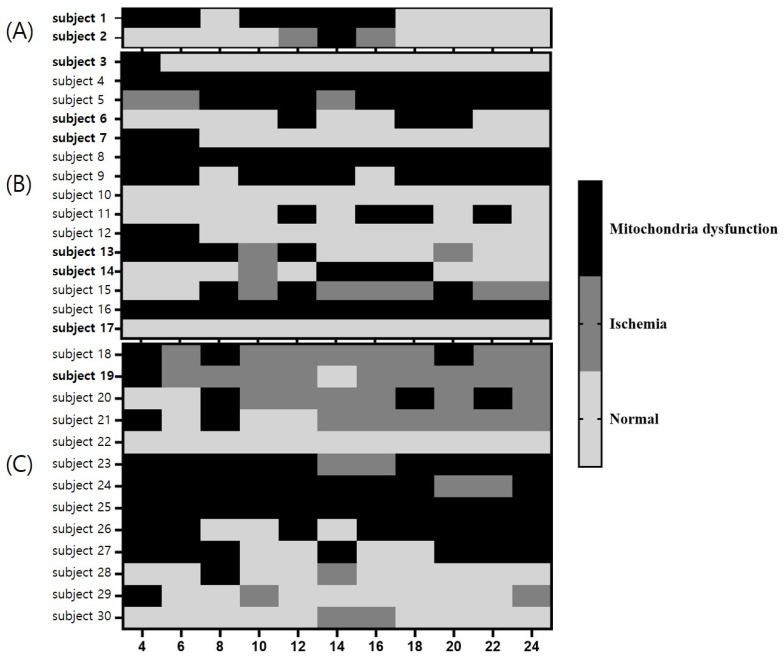
Distribution of cerebral ischemia and cerebral mitochondrial dysfunction in enrolled patients classified into low ((**A**) subjects 1 and 2), moderate ((**B**) subjects from 3 to 17), and high severity ((**C**) subjects from 18 to 30) groups based on the rCAST score. The subjects indicated in bold letters were patients with a good neurological prognosis. Patients with high-severity CA demonstrated a higher frequency of cerebral mitochondrial dysfunction or ischemia than those with moderate-severity. Abbreviations: rCAST, revised post-cardiac arrest syndrome for therapeutic hypothermia.

**Table 1 brainsci-13-01373-t001:** General patient characteristics.

Variable	Low Severity, n = 2	Moderate Severity, n = 15	High Severity, n = 13	*p* Value
Age, years	54.00 (45.00–)	60.00 (41.00–74.00)	50.00 (30.50–68.00)	0.16
Sex, male	1 (50.0)	11 (73.3)	8 (61.5)	0.70
Witnessed	2 (100.0)	11 (73.3)	4 (30.8)	0.03
Bystander CPR	2 (100.0)	7 (46.7)	11 (84.6)	0.06
Shockable rhythm	2 (100.0)	5 (33.3)	3 (23.1)	0.10
Cardiac etiology	2 (100.0)	5 (33.3)	4 (30.8)	0.16
Poor neurological prognosis	0 (0.0)	9 (60.0)	12 (92.3)	0.02
GCS 3 4 5 6				0.09
	1 (50.0)	13 (86.6)	13 (100.0)	
	0 (0.0)	0 (0.0)	0 (0.0)	
	0 (0.0)	0 (0.0)	0 (0.0)	
	1 (50.0)	1 (6.7)	0 (0.0)	
7	0 (0.0)	1 (6.7)	0 (0.0)	
8	0 (0.0)	0 (0.0)	0 (0.0)	
Induction time, h	4.63 (4.36–)	7.12 (4.53–8.20)	5.00 (4.45–6.29)	0.15
No flow time, min	0.50 (0.00–)	2.00 (0.00–14.75)	5.00 (0.50–18.00)	0.34
Low flow time, min	6.50 (6.00–)	23.50 (15.75–37.25)	24.00 (17.00–34.50)	0.83
Lumbar puncture time, h	3.54 (3.00–)	4.00 (2.00–4.00)	4.00 (3.00–4.45)	0.51
rCAST score	4.25 (3.00–)	10.50 (8.50–13.00)	17.50 (15.25–18.00)	<0.001
CCI	3.50 (1.00–)	4.00 (1.00–7.00)	1.00 (0.00–3.00)	0.08
Hypertension	0	7 (46.7)	3 (23.1)	
Diabetes mellitus	0	7 (46.7)	2 (15.4)	
Myocardial infarction	0	2 (13.3)	0	
Cerebrovascular attack	0	1 (6.7)	1 (7.7)	
Lung disease	0	2 (13.3)	2 (15.4)	
Renal disease	0	5 (33.3)	1 (7.7)	
Liver disease	0	0	0	
Malignancy	1 (50.0)	0	1 (7.7)	

Data are presented as median values (interquartile ranges) or the numbers (%). Abbreviations: CPR, cardiopulmonary resuscitation; GCS, Glasgow Coma Scale; CCI, Charlson Comorbidity Index.

**Table 2 brainsci-13-01373-t002:** Predictive value of CSF lactate levels for a poor neurological prognosis at 4 and 24 h after ROSC.

Time	Overall				Moderate-Severity Group			
	AUROC (95% CI)	Cut-Off Value	PPV	NPV	AUROC (95% CI)	Cut-Off Value	PPV	NPV
4 h	0.79 (0.44–0.97)	4.18	100.0	66.7	0.78 (0.45–0.96)	3.22	100.0	77.8
24 h	0.90 (0.72–0.98)	2.28	100.0	64.3	0.98 (0.72–1.00)	2.27	100.0	85.7

Abbreviations: AUROC, area under the receiver operating characteristic curve; CI, confidence interval; PPV, positive predictive value; NPV, negative predictive value; ROSC, return of spontaneous circulation.

**Table 3 brainsci-13-01373-t003:** Comparison of cerebral metabolism between the groups with moderate- and high-severity CA.

Time	Metabolism	Moderate Severity, n = 15	High Severity, n = 13	*p* Value
4 h after ROSC	Normal	6 (40.0)	4 (30.8)	0.81
	Cerebral MD	8 (53.3)	9 (69.2)	
	Cerebral ischemia	1 (6.7)	0 (0.0)	
24 h after ROSC	Normal	9 (60.0)	3 (23.1)	0.04
	Cerebral MD	5 (33.3)	5 (38.5)	
	Cerebral ischemia	1 (6.7)	5 (38.5)	

Abbreviations: MD, mitochondrial dysfunction; ROSC, return of spontaneous circulation.

## Data Availability

The data presented here are available on request from the corresponding author. The data are not publicly available because of ethical concerns.

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
