# Peer review of "Differences in Cerebral Metabolism between Moderate- and High-Severity Groups of Patients with Out-of-Hospital Cardiac Arrest Undergoing Target Temperature Management"

_brainsci, 2023, doi:10.3390/brainsci13101373_

Round 1

Reviewer 1 Report

Authors investigate the value of CSF lactate/pyruvate ratio (LP) in out hospital cardiac arrest in a single center. Authors conclude that the LP  car be use as a prognostic factor during 24h post ROC. 

Three major concerns are: 1- the low number of patients included, 2-single center included, 3-Monitoring CSF could be agressive but patients are not conscious.

Abstract: no comments

Introduction:  no comments

Methods: What means this sentence “next of kin refused lumbar punction…..”Line 89

Results: Table 3 should be available for the three severity populations.

Discussion: no comments

Limitations: no comments

Conclusion: no comments

Author Response

Comments from reviewer #1:

1,2> the low number of patients included, single center included,

Response: I agree that this study was conducted at a single center with a small sample size. However, in a previous study, the AUROC for CSF lactate levels measured 24 h after ROSC for the prediction of a poor neurological prognosis was 0.89, and 30 patients were required to achieve a power of 0.99 at a significant level of 0.05, considering a 10% missing rate. Additionally, a post hoc sample size calculation revealed that AUROC for CSF lactate levels measured 24 h after ROSC for prediction of a poor neurologic prognosis was 0.90, and that 22 patients were required to achieve a power of 0.99 at a significance level of 0.05. I have added this explanation in the limitations paragraph in the Discussion section.

3> Monitoring CSF could be aggressive but patients are not conscious.

Response: In the past decades, lumbar CSF drainage was contraindicated because of the risk of transtentorial or tonsillar herniation when the intracranial pressure increased. However, recent studies have shown that intracranial pressure rarely exceeds 20 mmHg in OHCA patients undergoing TTM, and that lumbar CSF drainage, when performed with careful monitoring of intracranial pressure and vital signs, might be a valuable treatment option for managing refractory increased intracranial pressure in patients with discernible basal cisterns. I have added this explanation in the Discussion section.

4> Methods: What means this sentence “next of kin refused lumbar punction…..”Line 89

Response: I apologize for the lack of clarity. We evaluated cerebral metabolism using CSF obtained through a lumbar catheter following lumbar puncture. Patients whose next of kin did not provide consent for lumbar puncture or other procedures were excluded from this study. Therefore, for better comprehension, the expression has been revised to “consent for lumbar puncture or other procedures not granted by the next of kin.”

5> Results: Table 3 should be available for the three severity populations.

Response: We aimed to assess the predictive value of CSF lactate for neurological prognosis across three severity groups. However, thirty-two patients with low severity demonstrated conscious recovery with a GCS score of ≥9 after ROSC. Only two patients with low severity were included in this study, and they exhibited good neurological prognosis. Among the 13 patients with high severity, only one demonstrated good neurological prognosis. Because of this distribution, a comparative analysis of the prognostic value of CSF lactate based on severity was not feasible. Therefore, the prognostic value of CSF lactate levels was compared between the overall patients and the moderate-severity group. I have presented these findings in the Results section.

Reviewer 2 Report

The submitted manuscript titled "Differences in cerebral metabolism based on severity in patients with out-of-hospital cardiac arrest undergoing target temperature management" by You et al. investigates the prognostic capabilities of parameters of cerebral metabolism in patients with out-of-hospital cardiac arrest who achieved spontaneous circulation in terms of their neurological outcome. Firstly, it is shown that more severe cases (as assessed using the rCAST score) have higher rates of longer lasting mytochondrial dysfunction (defined as high lactate/pyruvate ratio in cerebro-spinal fluid and CSF pyruvate level >0.07 mmol/L) and "normalization" by 24 hours after ROSC is more in favor of a good neurological prognosis (6 months after ROSC). Secondly, the analysis of lactate levels in the CSF shows a good positive predictive value to estimate neurological prognosis.

Overall, I am impressed with the quality of the article and believe that it makes a valuable contribution to the literature on this topic. In particular, I want to praise the clear structure of the article, the thorough and laborous data collection and analysis, and the overall quality of presentation. The weaknesses of the article are properly discussed.

The article is well-structured, with clear headings and subheadings that guide the reader through the content. In terms of language, the article is well-written and easy to follow. The authors use appropriate scientific terminology and provide clear definitions when necessary. The article is also free of spelling and syntax errors, which is a credit to the authors' attention to detail.

In conclusion, the manuscript at hand is a well-written and thoroughly researched article that provides valuable insights into the topic and offers an interesting and timely approach to assess prognosis in such patients.

However, there are a few remarks from my side that need to be addressed:

* methods: in section 2.2 it says "The patients’ neurological status was investigated by directly calling their caregivers 6 months after ROSC.", later in that section the CPC is mentioned (by the way: please add a reference for the CPC), but it is not clear, that the CPC is derived from the above call; only in section 2.4 it says "CPC at 6 months after ROSC"; please elaborate that part in sections 2.2 to make it more clear

* l. 151: "3 patients died after organ donation" - what does that mean exactly? I suppose organ donation was performed after brain death?

* l. 160: "The low-severity group exhibited a good neurological prognosis in all cases" - n = 2, the phrase seems a little presumptuous

* l. 161: "moderate-severity group had a poor neurological prognosis in 6 (60.0%) cases" --> 9 cases, cf. table 1

* observations from the presented data: no flow time correlates with poor neurological prognosis, inverse correlation of neurological prognosis and witnessed CA / shockable rhythm; that should be discussed

* in my opinion table 2 can be removed

* the presentation of different ROC analyses for the CSF lactate levels over 24h seems a little overkill and lacks practical usability; as it seems, 24h CSF lactate levels seem to have a good PPV in both groups; what CSF lactate levels indicate poor prognosis? please provide cut-off levels, ideally for the 24h sample

* sections 3.3 and 3.4 are hard to follow - maybe adding a table or figure would be helpful?

Author Response

Comments from reviewer #2:

1>  methods: in section 2.2 it says "The patients’ neurological status was investigated by directly calling their caregivers 6 months after ROSC.", later in that section the CPC is mentioned (by the way: please add a reference for the CPC), but it is not clear, that the CPC is derived from the above call; only in section 2.4 it says "CPC at 6 months after ROSC"; please elaborate that part in sections 2.2 to make it more clear

Response: I apologize for the lack of clarity. Neurological prognosis was assessed at 6 months after ROSC using the Glasgow–Pittsburgh Cerebral Performance Categories (CPC) scale, through either face-to-face interviews or structured telephone interviews [12], as follows: CPC 1 (good performance), CPC 2 (moderate disability), CPC 3 (severe disability), CPC 4 (vegetative state), or CPC 5 (brain death or death) [13]. Telephone interviews were conducted by an emergency physician who was fully informed of the protocol. I have added this explanation in the Materials and Methods section.  

2> l. 151: "3 patients died after organ donation" - what does that mean exactly? I suppose organ donation was performed after brain death?

Response: I apologize for the incorrect expression. I have revised the sentence to “…and organ donation was performed after brain death in 3 patients.”

3> l. 160: "The low-severity group exhibited a good neurological prognosis in all cases" - n = 2, the phrase seems a little presumptuous

Response: Thank you for pointing this out. To avoid unnecessary repetition, we have deleted this sentence and revised the sentence in the previous paragraph as follows:

Original: Throughout the research period, most patients with low severity were excluded from the study because they had a GCS score of ≥9. Only two patients with low severity were included in this study.

Revised: Throughout the research period, 32 patients with low severity were excluded from the study because they demonstrated conscious recovery with a GCS score of ≥9 after ROSC. Only two patients with low severity were included in this study.

The two patients with low severity exhibited good neurological prognosis.

4>  l. 161: "moderate-severity group had a poor neurological prognosis in 6 (60.0%) cases" --> 9 cases, cf. table 1

Response: I apologize for the error. I have revised this part of the sentence to “In the moderate-severity group, 9 (60.0%) cases showed a poor neurological prognosis,…”

5> observations from the presented data: no flow time correlates with poor neurological prognosis, inverse correlation of neurological prognosis and witnessed CA / shockable rhythm; that should be discussed

 Response: In previous studies, parameters such as shockable rhythm, witnessed CA, no flow time, low flow time, cardiac etiology, and age were associated with the neurological prognosis of OHCA patients. However, in this study, the majority of patients exhibiting low severity according to the rCAST score demonstrated conscious recovery after ROSC and were excluded from the study. Consequently, apart from witnessed CA, there were no significant between-group differences in other variables. I have added this explanation in the Discussion section.

6> in my opinion table 2 can be removed

Response: I have removed Table 2 per your recommendation.

7> the presentation of different ROC analyses for the CSF lactate levels over 24h seems a little overkill and lacks practical usability; as it seems, 24h CSF lactate levels seem to have a good PPV in both groups; what CSF lactate levels indicate poor prognosis? please provide cut-off levels, ideally for the 24h sample

Response: Thank you for pointing this out. I have provided the AUROC values for CSF lactate levels at 4 and 24 h after ROSC, along with the corresponding cut-off values, in Table 2.

8> sections 3.3 and 3.4 are hard to follow - maybe adding a table or figure would be helpful?

Response: I apologize for the lack of clarity. I have added Table 3 for better understanding of sections 3.3 and 3.4.

Reviewer 3 Report

I reviewed with interest the article by Yeonho You et al, “Differences in cerebral metabolism based on severity in patients with out-of-hospital cardiac arrest undergoing target temperature management.” In this article, the authors showed that the prognostic value of CSF lactate was better in the moderate group than in the overall patient population. Accordingly, the authors believe that the severity of a patient's condition should be taken into account when studying cerebral metabolism or using CSF lactate as a prognostic tool. These data may be useful in practice. However, while reviewing the manuscript, I had questions and comments to which I would like to receive answers from the authors.

1. Old sources – 17 out of 26 are older than 6 years, including 10 older than 10 years. For a current scientific article, most of the cited publications must be from the last 5 years. The following may be considered as possible publications (see 1-4 below).

2. Clinical characteristics are presented extremely sparingly. There is no information about the presence of comorbidity in the groups (the Charlson index is clearly insufficient), rCAST score values, etc.

3. The title of Table 3 (Predictive value of CSF lactate levels) is incomplete; it is unclear what CSF lactate levels predict.

4. A significant limitation is the large dropout of patients initially included in the study (30 of the original 153 were considered)

5. Figure 2 and its captions do not correspond to each other. The authors write: “Distribution of cerebral ischemia and cerebral mitochondrial dysfunction in enrolled patients classified into low- (A), moderate- (B), and high-severity (C) groups based on the rCAST score” (lines 232-233). However, there are no drawings under the letters A, B and C in Figure 2.

6. The need to separate a Low severity group from two patients is questionable. When analyzing, the authors do not compare the data in this group with groups with Moderate severity and High severity. Accordingly, it would be more correct to analyze these two groups, reflecting this in the title of the article.

7. The phrase “the prognostic value of CSF lactate was better in the moderateseverity group than in the overall patients” (in the abstract and conclusion) requires clarification, since it is unclear what kind of prognosis we are talking about.

References:

1.       Chen CH, Wang CJ, Wang IT, Yang SH, Wang YH, Lin CY. Does One Size Fit All? External Validation of the rCAST Score to Predict the Hospital Outcomes of Post-Cardiac Arrest Patients Receiving Targeted Temperature Management. J Clin Med. 2022 Dec 28;12(1):242. doi: 10.3390/jcm12010242.

2.       Kim N, Kitlen E, Garcia G, Khosla A, Miller PE, Johnson J, Wira C, Greer DM, Gilmore EJ, Beekman R. Validation of the rCAST score and comparison to the PCAC and FOUR scores for prognostication after out-of-hospital cardiac arrest. Resuscitation. 2023 Jul;188:109832. doi: 10.1016/j.resuscitation.2023.109832.

3.       Amagasa S, Yasuda H, Oishi T, Kodama S, Kashiura M, Moriya T. Target Temperature Management Following Pediatric Cardiac Arrest: A Systematic Review and Network Meta-Analysis to Compare the Effectiveness of the Length of Therapeutic Hypothermia. Cureus. 2022 Nov 18;14(11):e31636. doi: 10.7759/cureus.31636.

4.       Mishra SB, Patnaik R, Rath A, Samal S, Dash A, Nayak B. Targeted Temperature Management in Unconscious Survivors of Postcardiac Arrest: A Systematic Review and Meta-analysis of Randomized Controlled Trials. Indian J Crit Care Med. 2022 Summer;26(4):506-513. doi: 10.5005/jp-journals-10071-24173.

No comments

Author Response

Comments from reviewer #3:

  1. Old sources – 17 out of 26 are older than 6 years, including 10 older than 10 years. For a current scientific article, most of the cited publications must be from the last 5 years. The following may be considered as possible publications (see 1-4 below).

Response: Thank you for the excellent recommendations. I have now updated the references to include the latest relevant publications.

  1. Clinical characteristics are presented extremely sparingly. There is no information about the presence of comorbidity in the groups (the Charlson index is clearly insufficient), rCAST score values, etc.

Response: I apologize for the lack of information. I have now added details regarding comorbidities and the rCAST scores in Table 1.

Variables

Low severity,

n=2

Moderate severity,

n=15

High severity,

n=13

P value

Age, years

54.00 (45.00-  )

60.00 (41.00-74.00)

50.00 (30.50-68.00)

0.16

Sex, male

1 (50.0)

11 (73.3)

8 (61.5)

0.70

Witnessed

2 (100.0)

11 (73.3)

4 (30.8)

0.03

Bystander CPR

2 (100.0)

7 (46.7)

11 (84.6)

0.06

Shockable rhythm

2 (100.0)

5 (33.3)

3 (23.1)

0.10

Cardiac etiology

2 (100.0)

5 (33.3)

4 (30.8)

0.16

Poor neurological prognosis

0 (0.0)

9 (60.0)

12 (92.3)

0.02

GCS

3

4

5

6

0.09

1 (50.0)

13 (86.6)

13 (100.0)

0 (0.0)

0 (0.0)

0 (0.0)

0 (0.0)

0 (0.0)

0 (0.0)

1 (50.0)

1 (6.7)

0 (0.0)

7

0 (0.0)

1 (6.7)

0 (0.0)

8

0 (0.0)

0 (0.0)

0 (0.0)

Induction time, h

4.63 (4.36-  )

7.12 (4.53-8.20)

5.00 (4.45-6.29)

0.15

No flow time, min

0.50 (0.00-  )

2.00 (0.00-14.75)

5.00 (0.50-18.00)

0.34

Low flow time, min

6.50 (6.00-  )

23.50 (15.75-37.25)

24.00 (17.00-34.50)

0.83

Lumbar puncture time, h

3.54 (3.00-  )

4.00 (2.00-4.00)

4.00 (3.00-4.45)

0.51

rCAST score

4.25 (3.00-  )

10.50 (8.50-13.00)

17.50 (15.25-18.00)

<0.001

CCI

Hypertension

Diabetes mellitus

Myocardial infarction

Cerebrovascular attack

Lung disease

Renal disease

Liver disease

Malignancy

3.50 (1.00-  )

4.00 (1.00-7.00)

1.00 (0.00-3.00)

0.08

0

7 (46.7)

3 (23.1)

0

7 (46.7)

2 (15.4)

0

2 (13.3)

0

0

1 (6.7)

1 (7.7)

0

2 (13.3)

2 (15.4)

0

5 (33.3)

1 (7.7)

0

0

0

1 (50.0)

0

1 (7.7)

  1. The title of Table 3 (Predictive value of CSF lactate levels) is incomplete; it is unclear what CSF lactate levels predict.

Response: Thank you for pointing this out. I have revised the title of Table 2 as follows (Table 3 was changed to Table 2 as another reviewer recommended deleting Table 2): Predictive value of CSF lactate levels for poor neurological prognosis at 4 and 24 h after ROSC.

Table 2. The value of CSF lactate levels to predict the poor neurological prognosis at 4 and 24 h after ROSC.

Time

Overall

Moderate severity group

AUROC (95% CI)

Cut-off value

PPV

NPV

AUROC (95% CI)

Cut-off value

PPV

NPV

4 h

0.79 (0.44–0.97)

4.18

100.0

66.7

0.78 (0.45–0.96)

3.22

100.0

77.8

24 h

0.90 (0.72–0.98)

2.28

100.0

64.3

0.98 (0.72–1.00)

2.27

100.0

85.7

Abbreviations: AUROC, area under the receiver operating characteristic curve; CI, confidence interval; PPV, positive predictive value; NPV, negative predictive value; ROSC, return of spontaneous circulation.

  1. A significant limitation is the large dropout of patients initially included in the study (30 of the original 153 were considered)

Response: I agree that this study was conducted at a single center with a small sample size. However, in a previous study, the AUROC for CSF lactate levels measured 24 h after ROSC for the prediction of a poor neurological prognosis was 0.89, and 30 patients were required to achieve a power of 0.99 at a significant level of 0.05, considering a 10% missing rate. Additionally, a post hoc sample size calculation revealed that AUROC for CSF lactate levels measured 24 h after ROSC for prediction of a poor neurologic prognosis was 0.90, and tthat 22 patients were required to achieve a power of 0.99 at a significance level of 0.05. I have added this explanation in the Discussion section.

Line 284 – 290: However, in a previous study, the AUROC of CSF lactate levels measured 24 h after ROSC to predict poor neurologic prognosis was 0.89, and thirty patients were required to achieve a power of 0.99 at a significant level of 0.05, considering a 10% missing rate. Additionally, a post hoc sample size calculation revealed that the AUROC of CSF lactate levels measured 24 h after ROSC to predict poor neurologic prognosis was 0.90, and twenty-two patients were required to achieve a power of 0.99 at a significant level of 0.05.

  1. Figure 2 and its captions do not correspond to each other. The authors write: “Distribution of cerebral ischemia and cerebral mitochondrial dysfunction in enrolled patients classified into low- (A), moderate- (B), and high-severity (C) groups based on the rCAST score” (lines 232-233). However, there are no drawings under the letters A, B and C in Figure 2.

Response: I apologize for the inconvenience. I have now added the letters A, B, and C in Figure 2.

Figure 2. Distribution of cerebral ischemia and cerebral mitochondrial dysfunction in enrolled patients classified into low- (A; subject 1, 2), moderate- (B; subject from 3 to 17), and high-severity (C; subject from 18 to 30) groups based on the rCAST score. The subjects in bold letters were patients with a good neurological prognosis. Patients with high severity demonstrated a higher frequency of cerebral mitochondrial dysfunction or ischemia compared with those with moderate severity. Abbreviations: rCAST, revised post-cardiac arrest syndrome for therapeutic hypothermia.

  1. The need to separate a Low severity group from two patients is questionable. When analyzing, the authors do not compare the data in this group with groups with Moderate severity and High severity. Accordingly, it would be more correct to analyze these two groups, reflecting this in the title of the article.

Response: As shown in the Materials and Methods and Results sections, I excluded the group with low severity and compared the moderate- and high-severity groups. I have accordingly revised the title to “Differences in cerebral metabolism between moderate- and high-severity groups of patients with out-of-hospital cardiac arrest undergoing target temperature management”.

  1. The phrase “the prognostic value of CSF lactate was better in the moderate severity group than in the overall patients” (in the abstract and conclusion) requires clarification, since it is unclear what kind of prognosis we are talking about.

Response: I apologize for the lack of clarity. I have revised the sentence in the conclusions of the Abstract and main text to “The predictive value of CSF lactate levels for a poor neurological prognosis was better for patients with moderate severity than for the overall patient cohort.”

Line 178 – 182: Therefore, the prognostic value of CSF lactate levels was compared between the overall patients and the moderate-severity group. The value of CSF lactate in predicting poor neurological prognosis differed between the overall patients and the moderate-severity group, with the moderate-severity group demonstrating superior performance (Table 2).

Round 2

Reviewer 3 Report

The authors answered my questions and comments and made corrections to the manuscript. I have no other comments.

No comments